# Impact of Risk and Benefit on the Suppliers' and Managers' Intention of Shared Parking in Residential Areas

**Jin Xie [1], Xiaofei Ye [1,2,\*], Zhongzhen Yang [1], Xingchen Yan [3], Lili Lu [1,4], Zhen Yang [3] and Tao Wang [5]**

1   Faculty of Maritime and Transportation, Ningbo University, Ningbo 315211, China; xiejin0428@126.com (J.X.); yangzhongzhen@nbu.edu.cn (Z.Y.); lulili@nbu.edu.cn (L.L.)
2   Ningbo Port Trade Cooperation and Development Collaborative Innovation Center, Ningbo University, Ningbo 315211, China
3   College of Automobile and Traffic Engineering, Nanjing Forestry University, Nanjing 210037, China; xingchenyan.acad@gmail.com (X.Y.); zyang_2016@163.com (Z.Y.)
4   National Traffic Management Engineering & Technology Research Centre Ningbo University Sub-center, Ningbo University, Ningbo 315211, China
5   School of Architecture and Transportation, Guilin University of Electronic Technology, Guilin 541004, China; wangtao_seu@163.com
\*   Correspondence: yexiaofei@nbu.edu.cn; Tel.: +86-1526-785-9815

**Abstract:** Shared parking is not commonly applied in residential areas. The reason is that parking suppliers and managers believe that there are many uncertainties and conflicts in obtaining sharing benefits and taking sharing risks. To increase their acceptance of shared parking in residential areas, risk and benefit factors were identified by an influential analysis and a questionnaire survey. A research framework based on the structural equation model was developed to analyze the relationship between shared-parking risks, shared-parking benefits, management pressure, and intentions of parking suppliers and managers. The results showed that, to parking suppliers, the risks of shared parking have the largest effect on suppliers' intention to apply shared parking by a standardized coefficient of −0.85, followed by the benefits of shared parking (0.29), and management pressures (−0.14). To the parking managers, management pressures have the largest effect on managers' intention to apply shared parking by a standardized coefficient of −0.74, followed by the benefits of shared parking (0.52) and risks of shared parking (−0.46). These results can help in increasing parking suppliers' and managers' acceptance of shared parking in residential areas.

**Keywords:** transportation management; shared parking; stakeholders; risks and benefits; structural equation model

## 1. Introduction

Parking problems severely limit the sustainable development of cities in China. Shared parking has been applied to solve the problem of "parking difficulty". It has become a hot topic in the parking industry and in academic research [1–3]. In practice, some cities like Ningbo, Shanghai, Beijing, and Guangzhou have attempted to share parking berths between residential areas and adjacent commercial areas. However, the implementation of shared parking is poor. For example, only a few owners are willing to share their parking berths in the case in Shanghai, and others are unwilling to take the risk of sharing their parking berth without earning enough benefits. The shared parking mode has not been widely accepted. The important reasons are as follows: (1) Shared parking involves many

stakeholders, including parking suppliers, managers, shared parking platforms, and government departments, etc. Different parties have different interests. There are conflicts and contradictions in risk-taking and benefit distribution, which makes it difficult to promote the mode of shared parking; (2) The risk and benefit categories of shared parking are not clearly defined. It is difficult to meet the interests of all stakeholders. Taking the residential areas as an example, although parking suppliers who are the actual owners of parking berths can make extra income from outside vehicles using shared parking berths, they also suffer risks including the invasion of safety and privacy. Moreover, in theory, even though shared parking effectively solves the problem of parking demand and creates more social benefits, parking managers are under pressure to maintain the orders of the outside vehicles. Therefore, clarifying the risks and benefits for stakeholders is an important basis for the promotion of shared parking. It is of great value to better understand why people accept or reject shared parking.

This paper selects residential parking suppliers (the owners of parking berths in residential areas) and managers (the security personnel of parking lots) as stakeholders. The risk and benefit categories of stakeholders are further classified based on the interview survey method. Then, structural equation modeling is used to analyze the influence of the risks and benefits on the suppliers' and managers' intentions to apply the shared parking mode.

## 2. Literature Review

In a study of the influence of the urban parking pricing problem, Hensher and King [4] studied the impact of parking time limit policy and parking charge on parking choices. The results showed that parking charge had a significant impact on parking choice, with an influential rate of 97%, while the effect of time limit policy was only 3%. Rye [5] determined the structure of parking price by analyzing the influential factors of parking fee and studied how to make good use of these important factors through a reasonable parking fee to effectively improve the traffic environment. Taking different land use characteristics as the research object, Ison [6] summarized the main traffic problems in the central areas of British cities and studied how to improve traffic congestion through appropriate parking charge policies.

In a study of the pricing policies in urban contexts, Greg [7] stated that appropriate parking charges have no significant impact on the economic vitality of commercial districts. Chu et al. [8] stated that the management of parking spaces by a private company could maximize social benefits and realize marketization of parking fee management.

In a study of the user behavior in the case of different parking schemes, Hess [9] developed a polynomial Logit model to analyze and verify the impact of parking fee policy on commuter's choice of travel mode. The result indicated that, if parking were free, 62% of travelers would choose to drive themselves quickly and conveniently, 10% would choose to take a bus, and the rest would choose other public transportation options. Lambe [10] found that driving distance, parking fee, and walking distance after parking were closely related to parking choice.

In a study of the effects of looking for parking (cruising) on traffic flow, Ding et al. [11] built a parking behavior model by considering the influence of time value. The results showed that the search time of non-work trips was longer than that of work trips. Arnott [12] analyzed the change of parking search time from walking distance and parking space and established a model with a parking space occupancy rate of 85% as the optimal target. According to this research, activities in a specific period of time aggravate the generation of parking search behavior. Yan and Yang [13] proposed a time-benefit model of parking choice behavior to solve the theoretical problems of parking prediction and management in special activities. Arnott et al. [14] used the simple cellular automata condition simulation model to study the relationship between road occupancy rate and actual cruising time. Mannini [15] used FCD data (Floating Car Data) of the detection vehicle to identify cruising vehicles and model their cruising time. Zhao et al. [16] showed a station-oriented clustering analysis on ridership patterns in subway systems based on smart card date, and the results contributed to subway station ridership forecasting and provide theoretical basis for schedule making and adjustment.

Research on the management of shared parking began in the 1990s. Mary [17] put forward that shared parking is highly practical and there are no better operation or management modes to replace it at present. In the special report "Portland Urban Parking Sharing Mannual" made by Stein Engineering for Portland, OR, USA, the theory of parking berth sharing was applied to a typical case in practice. Based on the parking conditions in the Portland metropolitan area, this report established a parking lot sharing model and proposed that a parking lot sharing agreement should be signed between the plot according to the different opinions [18]. Another report in the Portland area listed eight land types, among which church land had the greatest potential for parking sharing, while office land, school, restaurant, and cinema land had the smallest potential for parking sharing [19]. Yu [20] proposed a two-layer decision model of parking sharing measures based on parking berth allocation simulation in 2011. In practice, the United States paid attention to publicizing the mode of shared parking before the implementation of parking sharing, so as to improve the public's awareness of the parking sharing strategy and the implementation effect of the sharing measures. Moreover, Ni [21] discussed the possible land types applicable to shared parking. The result showed that shared parking can be applied to movie theaters, restaurants, hotels, etc. Resha [22] studied the situation of shared parking in Portland, Oregon, through a large number of surveys on the users of shared parking; the key problems and obstacles affecting the efficiency of shared parking were concluded, and the applicable implementation regulations of shared parking were provided for the judiciary. Abdul et al. [23] combined with space technology and applied geographic information systems to establish the model of supply and demand of parking spaces. Taking Johor Baru as a case, they analyzed and developed the "SPATT model" to analyze the spatial distribution of the supply and demand of parking spaces in the research area. Fei [24] concluded that parking in residential areas is regular, and their study showed that the parking rate of residential land presented a stable change trend within 10% of the number of parking spaces in the working period and rest day; Zhen [25] established a two-level programming induction model based on shared parking in the residential area, it was used to measure whether the guidance service can realize the balanced utilization of regional parking resources and whether shared parking was feasible; Jian [26] established the optimal allocation model of shared berth resources in residential areas. Ommeren [27], Inga [28], and others proposed the problems of parking lot utilization and parking space sharing in residential areas, but did not involve specific research on sharing implementation.

The theory of shared parking has been studied in detail, but the researchers on shared parking in residential areas mainly focused on the analysis of the utilization characteristics of residential parking spaces. There is little study on the analysis of the risks and benefits of shared parking. Li [29] analyzed the influential factors of private car owners' intention to share parking and the influential factors of parking demanders to choose the shared parking mode. However, analysis of other risks and benefits such as social benefit, cost risk, management pressures of parking suppliers (owners), and managers were not brought forward. It is of great value to analyze parking suppliers and parking managers' risks and benefits of shared parking to put forward reasonable solutions to meet their interests for shared parking, and then accelerate the promotion of the shared parking model in residential areas.

## 3. Variables and Data

### 3.1. Questionnaire and Explanatory Variables

The interview survey was conducted in 12 residential areas in different districts in Ningbo. The suppliers and managers were asked to answer what they considered to be the risks and benefits of shared parking. According to the interview results, both the suppliers and managers agreed that shared parking provides more available parking resources and alleviates the parking difficulty in the residential areas. They could get benefits from the parking charges, but they also worried about the threats of safety and privacy invasion. Therefore, the risks and benefits of shared parking affect the acceptance intention of parking suppliers and managers. The interview results showed the main

benefits were from the economic and social benefits, and the main risks were from the increasing costs, the uncontrollable security threats and the increasing management pressures. All of the risks and benefits were defined as the latent variables. Since they could not be measured directly, proper multiple observed indicator variables must be used to define them.

In order to guarantee that the observed indicator variables are reasonable, they are selected based on the details of interview survey results, according to existing literatures about shared parking evaluation. Observed indicator variables for each latent variable are detailed in Table 1.

As shown in Table 1, for parking suppliers, the benefits of shared parking are divided into economic benefits and social benefits.

### Economic benefits

Parking demanders pay for parking fees when they use shared berths in residential areas, and the suppliers can gain economic income directly. The higher the income is, the higher the intention of suppliers to share parking berths. Therefore, "Gain income through shared parking charge directly" is taken as the measurable variable of economic benefits to study the intention of parking suppliers to share parking in the residential areas.

### Social benefits

Shared parking mode could break the restriction of the parking permit and provide more available berths for the vehicles. It is proposed to promote the sustainable development of the city by solving the problem of parking difficulties. Social benefits could be represented by the four contents of "Increase utilization of vacant berths, Improve the satisfaction of parking, Indirectly improve people's quality of life, Provide new impetus for sustainable development of the city". For the parking suppliers, the more social benefits are made, the higher the intention to share parking berths.

Similarly, for parking suppliers, the risks of shared parking are divided into cost risks, security risks, and management pressures.

### Cost risks

Cost risks are mainly the cost of the purchase of parking equipment and parking berth rebuilding. The higher the cost is, the lower the intention of parking suppliers to share parking spaces, so this paper takes the "Increase the cost for new equipment purchase and parking lot rebuilding, increase managers' salaries" as the measurement variables of cost risks to analyze parking suppliers' intention to shared parking spaces.

### Security risks

Meanwhile, traffic safety, privacy safety, and parking order are the major concerns of residents regarding shared parking mode. For the parking suppliers, the higher security risks brought by shared parking, the lower the intention of parking suppliers to share parking spaces. In order to describe the safety problems faced by residents after a shared parking application, this paper takes the two contents of "Residents' traffic safety is not guaranteed, Residents' privacy safety is not guaranteed" as the measurable variables of security risks to set questionnaire problems, and analyzes the effect of two factors on the suppliers' intention to share parking.

### Management pressures

The management pressures are generated from the invasions of traffic safety and personal privacy. They can be summarized in the following three aspects: increase the work of handling parking conflicts, increase the work of supervising outside vehicles, and increase the work of dealing with traffic accidents. In order to analyze the influence of management pressures on the intention to share parking, these three contents are defined as the observable variables of management pressures.

**Table 1.** Explanatory Variables from the risks and benefits of shared parking.

| Latent Variables | | Observable Variable | Serial Number |
|---|---|---|---|
| **Parking suppliers** | Benefits of Shared parking | Economic benefits | Gain the income through shared parking charge directly | DB0 |
| | | Social benefits | Increase the utilization of vacant berths | DB1 |
| | | | Improve the satisfaction of parking | DB2 |
| | | | Indirectly improve people's quality of life | DB3 |
| | | | Provide new impetus for sustainable development of city | DB4 |
| | Risks of Shared parking | Cost risks | Increase the cost for new equipment purchase and parking lot rebuilding | DC1 |
| | | | Increase the management salaries | DC2 |
| | | Security risks | Residents' traffic safeties are not guaranteed | DS1 |
| | | | Residents' privacy safeties are not guaranteed | DS2 |
| | | Management pressure | Increase the work of handling parking conflicts | DM1 |
| | | | Increase the work of supervising outside vehicles | DM2 |
| | | | Increase the work of dealing with traffic accidents | DM3 |
| | Suppliers' Intention | | Intention of the suppliers to apply shared parking in the residential areas | A1 |
| **Parking manager** | Benefits of Shared parking | Economic benefits | Increase managers' salaries | DF1 |
| | | Social benefits | Increase utilization of vacant berths | SG1 |
| | | | Improve parking satisfaction | SG2 |
| | | | Indirectly improve people's quality of life | SG3 |
| | | | Provide new impetus for sustainable development of urban | SG4 |
| | Risks of Shared parking | Security risk | Residents' traffic safety are not guaranteed | DG1 |
| | | | Residents' privacy are not guaranteed | DG2 |
| | | Management pressure | Increase the work of handling parking conflicts | MG1 |
| | | | Increase the work of supervising outside vehicles | MG2 |
| | | | Increase the work of dealing with traffic accidents | MG3 |
| | Managers' intention | | Intention of the managers to apply shared parking in the residential areas | A2 |

For the parking managers, the sharing benefits are also divided into economic benefits and social benefits, and the shared risks are divided into cost risk, safety risks, and management pressures. As shown in Table 1, the difference between observable variables of parking suppliers and parking managers are increased managers' salaries due to the cost risk for the parking suppliers but economic benefits for the parking managers. In general, the more pressures of management work there are, the lower the intention of parking managers to share parking spaces in the residential areas. There are also not any cost risks to the parking managers. Therefore, "Increase the work of handling parking conflicts, Increase the work of supervising outside vehicles, Increase the work of dealing with traffic accidents" are taken as the observable variables of social benefits, and "Residents' traffic safeties are not guaranteed, Residents' privacy safeties are not guaranteed" are taken as the measurable variables of safety risks. Many management works will be increased after shared parking application, such as the traffic management, safety management, and conflict resolution between residents and outside parking demanders, etc. With the increase of management pressures, the intention of managers to share parking spaces decreases. Thus, "Increase the work of handling parking conflicts, Increased supervision of vehicles from outside by managers, Increase the work of traffic safety management" are taken as the observable variables of management pressures to analyze the intention of parking managers.

*3.2. Questionnaire and Data Collection*

SP (State Preferences) survey is a method to investigate traveler's awareness of alternative options under assumed conditions. The purpose of the SP is to investigate the changes in people's thinking, consciousness, and actions and analyze the demand for non-existing service systems. The main feature of SP is that the investigated situations have not yet happened. However, RP (Revealed Preferences) survey method is developed to investigate the policies that have been implemented. The informants need to fill in the questionnaire according to their actual travel behavior to obtain probability of the actual use. The main characteristic of RP survey is that questionnaire content has already happened. It can be seen that the change of people's thinking, consciousness, and action can be grasped by investigating an event that has not happened yet. Therefore, this paper is more suitable for using SP survey method to conduct the questionnaire survey to obtain effective data. The difference between the survey method in this paper and the existing research is mainly the survey content. First, the objects of the survey are different. This paper focuses on the parking suppliers and parking managers in residential areas. Second, the content of the survey includes various risks, such as cost and safety risk. However, the existing studies did not analyze the cost risk and safety risk.

The questionnaire survey was designed according to the latent variables and the observable variables mentioned above. The questionnaire comprises two sections. The first section is about parking suppliers' intention to share parking, and the second section is about parking managers' intention to share parking. Eighteen questions are set for each section, including sociodemographic characteristics and relevant contents of observable variables. Sociodemographic characteristics include gender, age, occupation, vehicle ownership, etc. The observed variables are presented with a question and the respondents choose one answer from five options: Strongly agree, agree, generally disagree, and strongly disagree about whether each observation variable will affect the sharing intention of parking suppliers and parking managers according to their own situation. Through the internet and field survey, data collection of shared parking intention was carried out. After on-site observation and investigation, we chose residential areas with high parking demand from different CBD (Central Business District) areas in different districts. Three or four residential areas in the Haishu, Jiangbei, Yinzhou, and Beilun districts were observed. Then, the pilot survey was conducted to revise the questionnaires in these selected areas. The formal investigation was conducted randomly in other areas of the CBD. The field survey was conducted in Gulou, Chenghuang temple, Yuehu Shengyuan, and the Yinzhou district of Ningbo city for four days. Finally, 820 questionnaires were completed. Of the 820 questionnaires, 798 (valid response which included parking suppliers (498) and managers (300) in residential areas) valid ones were used in the following analysis. To maintain the

accuracy of the estimations and proper solutions, ensure representativeness, and use multiple observed indicator variables to define latent variables, a much larger and sufficient sample size, from 100 to 200, is recommended when maximum likelihood estimation is used. According to the study, a sample size of 798 is adequate for SEM.

Among the 798 valid questionnaires, for the parking suppliers, 51.19% of respondents were male and 48.81% were female. There are about the same number of males and females, which makes it reasonable to analyze the intention to share parking. For the parking managers, 87.56% of respondents were male and 12.44% were female. The number of male managers is far more than female managers, which reflects the fact that the number of male parking managers is actually more than female parking managers generally. More details are shown in Table 2.

The questions about the risks and benefits employ a five-point the Likert scale format [30] to score the benefits from "strong agreement" to "strong disagreement" with 5,4,3,2,1, and score the risks from "strong agreement" to "strong disagreement" with −5, −4, −3, −2, −1.

Based on the description above, the following hypotheses are obtained:

**H$_1$.** *The benefits positively influence the intention of parking suppliers to participate in shared parking application significantly.*

**H$_2$.** *The risks negatively influence the intention of parking suppliers to participate in shared parking practices significantly.*

**H$_3$.** *Management pressures positively influence the intention of parking suppliers to participate in shared parking significantly.*

**H$_4$.** *The benefits positively influence parking managers' intention to share parking spaces significantly.*

**H$_5$.** *The risks negatively influence parking managers' intention to share parking spaces significantly.*

**H$_6$.** *Management pressures negatively influence parking managers' intention to share parking spaces significantly.*

**H$_6$a.** *The benefits and risks of shared parking affect each other significantly.*

**H$_6$b.** *Management pressures and risks of shared parking affect each other significantly.*

**Table 2.** Characteristics of Respondents.

| Variable | | Category | Percent |
|---|---|---|---|
| **Parking suppliers** | Gender | Male | 51.19 |
| | | Female | 48.81 |
| | Age (years) | 18–25 | 32.12 |
| | | 26–35 | 48.35 |
| | | 36–45 | 13.48 |
| | | >45 | 6.05 |
| | Whether you use the parking space? | Yes | 69.05 |
| | | Sometimes | 17.86 |
| | | No | 13.1 |
| | How long will you be parking outside in the workday? (one day) | <6 h | 46.43 |
| | | 6-8 h | 32.14 |
| | | >8 h | 21.43 |
| | How long will you be parking at home in the non-workday? (one day) | <8 h | 45.24 |
| | | 8–12 h | 39.29 |
| | | >12 h | 15.48 |
| | Whether you want to apply the shared parking APP? | Yes | 70.24 |
| | | Do not care | 29.76 |

**Table 2.** *Cont.*

| Variable | | Category | Percent |
|---|---|---|---|
| **Parking managers** | Gender | Male | 87.56 |
| | | Female | 12.44 |
| | Age (years) | 18–25 | 2.38 |
| | | 26–35 | 3.57 |
| | | 36–45 | 59.52 |
| | | >45 | 34.52 |
| | How many parking berths are there in your parking spaces? | 1–50 | 44.52 |
| | | 51–100 | 25.11 |
| | | 101–150 | 18.33 |
| | | >150 | 12.04 |
| | How is usage of your parking space during the day? | Congested | 40.95 |
| | | Sufficient | 39.86 |
| | | Idle | 19.19 |
| | How is usage of your parking space during the night? | Congested | 50.24 |
| | | Sufficient | 35 |
| | | Idle | 14.76 |
| | Are you willing to share parking spaces? | Yes | 58.71 |
| | | Do not care | 22.47 |
| | | No | 18.82 |

## 4. Methods

This study was aimed at the influence of risks and benefits on the intention to share parking space. The latent variables and observable variables were shown in Table 2. The Structural Equation Model (SEM) was applied because of the complex relationships among these variables and their measurement error.

SEM methodology [31] can simultaneously analyze and capture the complex interrelationships among the intention to share parking spaces, economic benefits, social benefits, security risks, and management pressures. The effects of observed and latent variables can be decomposed into direct and indirect effects. SEM also allows a user to have standardized parameters that show the relative influences of observed and latent variables.

Figure 1 represents a set of recursive structural equations among the variables. In SEM, the underlying theory of the phenomena under investigation plays a key role in assessing model adequacy and testing relationships among the variables. A set of 8 independent exogenous variables was identified. These variables are the personal and parking characteristics that may influence the shared intention of parking suppliers and mangers, as discussed earlier in Section 3. BD0 & DF1, DB1-4 & SG1-4, DS1-2 & DG1-2, and DM1-3&MG1-3, were listed as indicators of the latent construct for economic and social benefits, security risks, management pressures respectively.

In addition, e1–e27 were the errors of each observable variable. Since the economic benefits can compensate for the cost in the shared parking, double arrows were used to represent the relationships between benefits and risks that affect each other. Similarly, management pressures and shared risks were linked by two-way arrows.

The conceptual SEM and measurement in this study can be represented by the following matrix equations [32]:

An equation relating the latent endogenous variable ($\eta_{en}$) to the latent exogenous variable ($\eta_{ex}$) and exogenous independent variables are as follows:

$$\eta_{en} = \beta\eta_{ex} + \Gamma_{en}X_{en} + \zeta_{en} \tag{1}$$

where:

$\eta$ = column vector of a latent variable;

$\beta$ = matrix of structural coefficients from the latent exogenous variable (*ex*) to the latent endogenous variable (*en*);

$\Gamma$ = matrix of structural coefficients from observed independent variables (*X*) to the latent endogenous variables (*en*) and latent exogenous variables (*ex*);

$\zeta$ = column vector of error terms associated with latent variable.

Equations relating the observed indicators (*Y*) to the latent constructs $\eta_{en}$ and $\eta_{ex}$ are as follows:

$$Y_{enk} = \Lambda_Y\eta_{en} + \varepsilon_{enk} \tag{2}$$

$$Y_{exk} = \Lambda_Y\eta_{ex} + \varepsilon_{exk} \tag{3}$$

where

*Y* = column vector of observed variables measured as deviations from their means;

*k* = number of observed indicator variables;

$\Lambda_Y$ = matrix of structural coefficients from the latent variables to their observed indicators;

$\varepsilon$ = column vector of measurement error terms of the observed indicator variables of the latent variables.

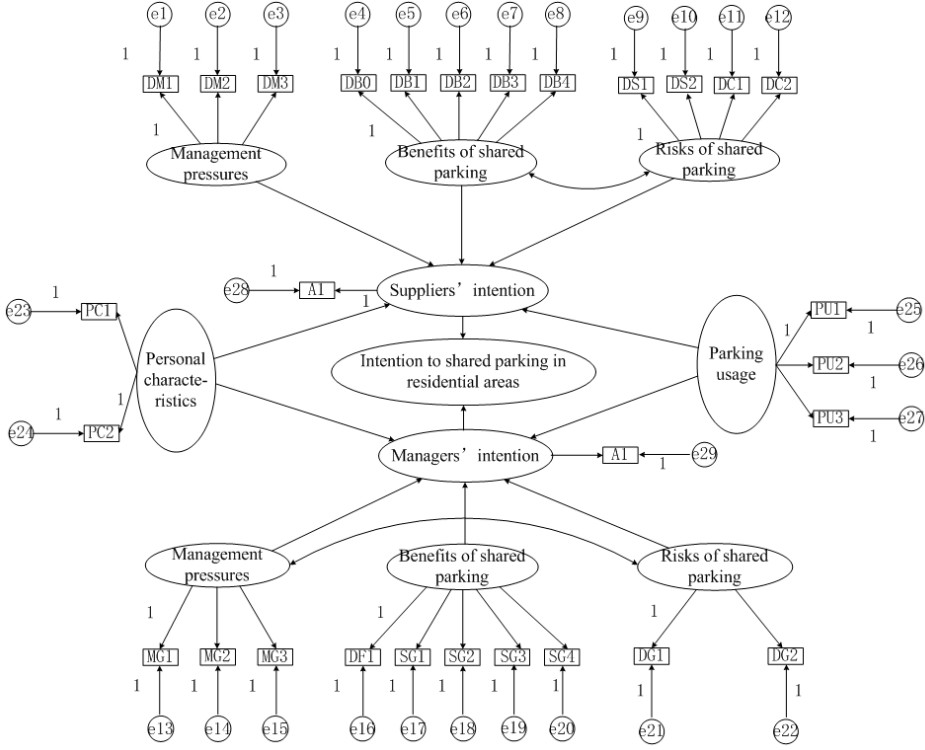

**Figure 1.** Path analysis model of the shared intention of parking suppliers and managers.

## 5. Goodness of Fit and Estimated Result

The initial model was gradually modified to an accepted model. Figure 2 shows the path diagrams according to the estimate results output in AMOS 21.0 software. Compared with the initial model, the final model in Figure 2 deletes two relationships—that between the intention and personal characteristics, and parking usage. The standardized factor loading and standardized path coefficients between latent variables are all shown in Figure 2. These standardized coefficients between different variables represent the relations' degree of strength.

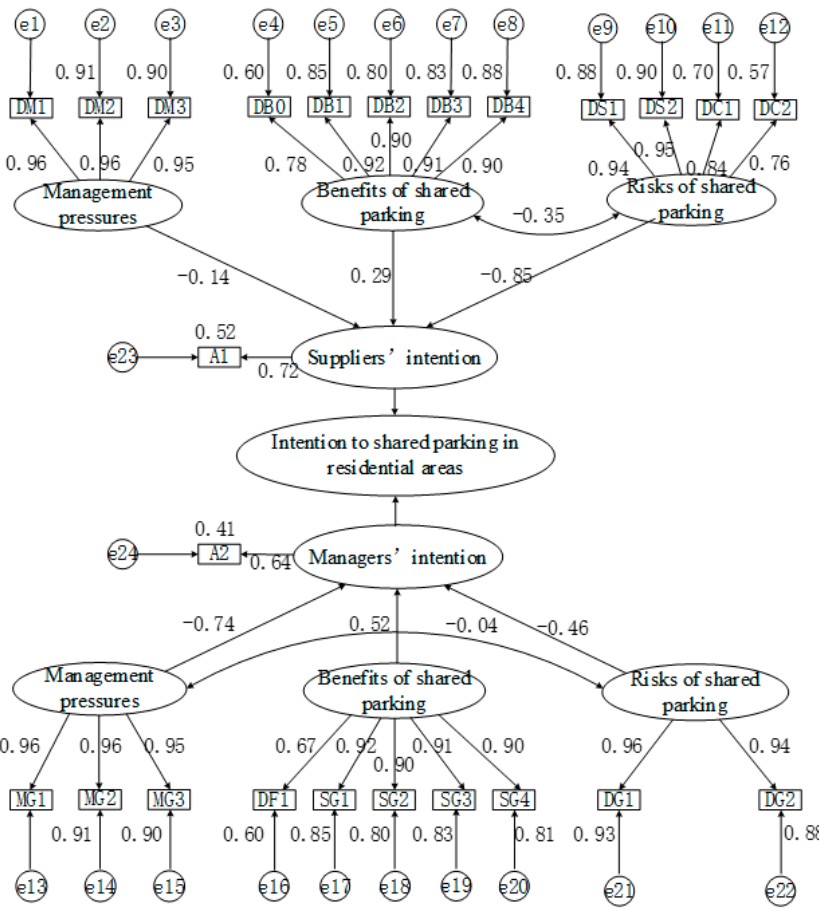

**Figure 2.** Measurement and structural model with standardized estimates.

The indices for goodness of fit are summarized in Table 3. The comparison of the absolute fix index with accepted criteria shows that $\chi^2$/degrees of freedom, goodness-of-fit index, and root-mean-square error of approximation all meet the requirements. In addition, since $\chi^2$/degrees of freedom <2 and root-mean-square error of approximation <0.06, the final model fits very well. The indices normed fit index = 0.974 > 0.9 and goodness-of-fit index = 0.949 > 0.9 conform with the requirements. Other indices, such as incremental fit index, 0.974; comparative fit index, 0.986; are higher than the accepted criterion of 0.9. All these indices indicate that the explanatory power of the model is high.

### 5.1. Hypothesis Testing

The overall fit indices show that the final model fits the data very well and is accepted. Hence, the hypothesis relationships in the conceptual framework can be tested through the standardized path coefficients between latent variables. The testing results for the eight assumed relationships are summarized in Table 4.

**Table 3.** Overall fit indices for model.

| Index | Value of Index | Criteria Value |
|---|---|---|
| $\chi^2$ | 135.909 | Does not exist |
| CMIN/DF | 2.192 | <5.0 |
| GFI | 0.949 | >0.9 |
| NFI | 0.974 | >0.9 |
| IFI | 0.986 | >0.9 |
| CFI | 0.986 | >0.9 |
| RMSEA | 0.055 | <0.1 |

NOTE: DF = degrees of freedom, GFI = goodness-of-fit index, NFI = normed fit index, IFI = incremental fit index, CFI = comparative fit index, RMSEA = root-mean-square error of approximation.

From Table 4, six of the eight hypotheses in the conceptual framework are significantly supported. The supported $H_1$ and $H_4$ indicate that the benefits of shared parking ($\beta_1 = 0.157$, $\beta_4 = 0.365$, $P_1 < 0.001$, $P_4 < 0.001$) positively and significantly influence the intention of the suppliers and mangers to apply shared parking. Conversely, the supported $H_2$ and $H_5$ indicate that the risks of shared parking ($\beta_1 = -0.360$, $\beta_4 = -0.187$, $P_1 < 0.001$, $P_4 < 0.001$) negatively and significantly influence the intention of the suppliers and mangers to apply shared parking. Furthermore, the relationships between management pressures and mangers' intention of shared parking are also significantly supported, which indicate that the management pressures ($\beta_6 = -0.323$, $P_6 < 0.001$) negatively and significantly influences the intention of the managers to apply shared parking. Moreover, the hypotheses $H_3$ was not significant, which indicates that the management pressures have no influence on the intention of the suppliers to share parking spaces. The reason is that the suppliers just provide parking spaces and are not involved in the management works. Compared to $H_3$, the supported $H_6$ means that the management works of shared parking may totally be accomplished by the managers. The higher the management pressures are, the lower the intention to share parking spaces.

**Table 4.** Hypothesis Testing Results.

| Hypothesis | Model Path | Estimate | C.R. | P | Test Result |
|---|---|---|---|---|---|
| $H_1$ | Benefits of Shared parking —> Suppliers' intention | $\beta_1 = 0.157$ | 5.205 | *** | significant |
| $H_2$ | Risks of Shared parking —> Suppliers' intention | $\beta_2 = -0.360$ | −15.300 | *** | significant |
| $H_3$ | Management pressures—> Suppliers' intention | $\beta_3 = -0.062$ | −2.865 | 0.204 | non-significant |
| $H_6a$ | Benefits of Shared parking <—> Risks of Shared parking | $\beta_7 = -0.457$ | −6.132 | *** | significant |
| $H_4$ | Benefits of Shared parking —> Managers' intention | $\beta_4 = 0.365$ | 7.343 | *** | significant |
| $H_5$ | Risks of Shared parking —> Managers' intention | $\beta_5 = -0.187$ | −7.070 | *** | significant |
| $H_6$ | Management pressures —> Managers' intention | $\beta_6 = -0.323$ | −11.728 | *** | significant |
| $H_6b$ | Management pressures <—> Risks of Shared parking | $\beta_8 = -0.063$ | −0.745 | 0.456 | non-significant |

Note: *** indicates a significant level of 0.1%.

Additionally, the hypotheses $H_6a$ was significantly supported, which indicates that the compensation effect of the benefits on the risks exists. The greater the benefits of shared parking, the stronger the risk tolerance. Unexpectedly, the hypotheses $H_6b$ ($P_6 = 0.456$) was not significant, which indicates that the management pressures have no influence on the risks of shared parking when the suppliers and managers consider whether the parking space in residential areas should be shared. The reason is that the managers should always focus on the parking management works throughout the application of shared parking and could not reduce their management pressures by evaluating the risks. In other words, whether the risk events happen or not, the management works and pressures for keeping parking orders always exist.

*5.2. Discussion of Results*

For details, more influential factors on the intention of the suppliers and managers to share parking spaces are analyzed as follows:

5.2.1. The Factors Influence on Parking Suppliers' Intention to Share Parking Spaces

(1) The path coefficient between the risks of shared parking and the suppliers' intention is −0.85, with the largest proportion among the influential factors. This result reflects that cost risks and security risks were the most important factors for parking suppliers, and the proportion of security risks was larger than cost risks. The path coefficients of the two observation variables "residents' traffic safety cannot be guaranteed (DS1)" and "residents' privacy security cannot be guaranteed (DS2)" are 0.94 and 0.95, respectively. Compared to cost risks, parking suppliers seemed to be more concerned with the safety of the residential areas after sharing parking spaces but were not concerned about cost. Therefore, the countermeasures that strengthen the security of residents should be taken during the shared parking application.

(2) The path coefficient between the benefits and risks of shared parking was −0.35. The two variables affect each other significantly. In the reality of shared parking application, the risks outweigh the benefits. This is the reason why shared parking is hard to apply commonly in residential areas. Hence, the countermeasures that apply shared parking mode widely should be taken to compensate for the risks with the benefits, or at least balance the risks and benefits. Moreover, a subsidy from the government department should be provided to encourage shared parking.

(3) As shown in Figure 2, the path coefficients of DB1–DB4 were 0.92, 0.90, 0.91, and 0.90, respectively, while DB0's was only 0.78. It indicates that the income brought by shared parking was not the main factor that influenced whether parking suppliers were willing to share parking spaces. For suppliers, whether or not social values were accomplished is the main factor for shared parking application. To promote suppliers' intention to participate in shared parking, a countermeasure is necessary to emphasize the social values and enhance the public awareness of social benefits.

5.2.2. The Factors' Influence on Parking Managers' Intention to Share Parking Spaces

(1) The path coefficient between management pressures and managers' intention was −0.74, with the largest proportion among the influential factors. The path coefficients of the observable variables of management pressures (MG1–MG3) were 0.96, 0.96, and 0.95, respectively. In order to improve the parking managers' intention in shared parking mode, it is necessary to reduce the pressures of the managers by increasing the salary and establishing a new supervision mechanism.

(2) Similar to parking suppliers, the four observed variables of social benefits create more contributions than the income through parking charges. As shown in Figure 2, the path coefficients of SG1–SG4 were 0.92, 0.90, 0.91, and 0.90, respectively, while the path coefficients of DF1 were 0.67. It can be seen that the increased management salary was not the main factor that determines whether parking managers participate in shared parking application, but the main concern was to make contributions to social development. Therefore, it is necessary to emphasize the benefits brought by shared parking to society, and then improve parking managers' intention to share parking in residential areas.

(3) The path coefficients of DG1 and DG2 to the shared risks were 0.96 and 0.94, respectively. It indicates that shared risks have an important influence on parking managers to participate in shared parking in residential areas. Therefore, policy support can be used to assist parking managers in parking management and reduce their pressures.

To sum up, management pressure was the most important factor for parking managers, followed by shared benefits and shared risks. The influence of management pressures outweighed the impact of shared benefits. Therefore, in order to improve the parking managers' intention, it is necessary to reduce the parking managers' management pressures or balance the managers' sense of pressure, such as by improving the sharing benefits.

## 5.3. Analysis of Direct, Indirect, and Total Effects

The direct effects and total effects between latent variables can be used to analyze the strength of each causal relationship. A direct effect is the influence of one variable on another that is not mediated by any other variables, and an indirect effect is one that is mediated by at least one other variable. The total effect of one variable on another is the sum of the direct and indirect effects. The path coefficients are shown in the previous subsection are all direct effects. Since an indirect relationship might exist between latent variables, it is often useful to calculate the direct and indirect effects from the model to get a better understanding of the model estimation results.

Direct, indirect, and total effects between latent variables are given in Table 5. They can be used to analyze the different weights of factors on acceptance intention. From Table 5, it can be seen that in two factors that are significantly related to parking suppliers' intention, risks of shared parking had the largest total effect by a coefficient of −0.85, followed by benefits of shared parking, which had a total effect on parking suppliers' intention by a coefficient of 0.29. Management pressures are non-significantly related to parking suppliers' intention with a coefficient of −0.14. In addition, there are three factors that were significantly related to parking managers' intention: management pressures had the largest total effect by a coefficient of −0.74, followed by benefits of shared parking and risks of shared parking, which had a total effect on managers' intention by coefficients of 0.52 and −0.46, respectively. It can also be seen that there were no indirect relationships among the six determinants of parking suppliers' intention and parking managers' intention.

**Table 5.** Direct, Indirect, and Total Effects Between Latent Variables.

| Relation Between Latent Variables | Direct Effects | Indirect Effects | Total Effects |
|---|---|---|---|
| Benefits of Shared parking —> Suppliers' Intention | 0.29 *** | — | 0.29 *** |
| Risks of Shared parking —> Suppliers' Intention | −0.85 *** | — | −0.85 *** |
| Management pressures—> Suppliers' Intention | −0.14 | — | −0.14 |
| Benefits of Shared parking <—> Risks of Shared parking | −0.35 *** | — | −0.35 *** |
| Benefits of Shared parking —> Managers' intention | 0.52 *** | — | 0.52 *** |
| Risks of Shared parking —> Managers' intention | −0.46 *** | — | −0.46 *** |
| Management pressures —> Managers' intention | −0.74 *** | — | −0.74 *** |
| Management pressures <—> Risks of Shared parking | −0.04 | — | −0.04 |

Note: *** indicates a significant level of 0.1%.

## 6. Discussion

From the SEM methodology results, the influential factors on the intention to share parking spaces were the risks, benefits, and management pressures. In order to apply the shared parking mode widely, implementation strategies that decrease the risks, increase the benefits, and balance the management pressures should be further proposed as follows.

Firstly, a new-style cooperative safety supervision mechanism of shared parking should be established by the stakeholders to reduce the security risks and cost risks. The supervision mechanism should be dominated by the government and cooperate with other stakeholders in China. For instance, the security risks mentioned in the variables DS1 and DS2. Residential safety and privacy should be strengthened by a supervision mechanism from governmental and operational parties. If the residents' safety and privacy were invaded by the outside parking vehicles, punishments like restriction of parking and violation of personal credit records should be carried out. The relevant policies of punishments and rewards should be regulated by the government.

Secondly, Advanced Intelligent Parking Technology (AIPT) should be introduced to reduce the pressures of shared parking management. The number of management staff will be decreased by the new AIPT, and the additional salary costs of managers will be saved, too. AIPT also provides the functions of monitoring, management, and control for each parking berth, so the problems caused by the influential factors MG1, MG2, and MG3 could be also solved by the AIPT.

Thirdly, demonstration projects of shared parking should be implemented by the government, supplier, and manager. The implementation effect and social values of shared parking should be broadcasted to the public. Through the demonstration projects, the benefits and the risks will be clearer in practice, so anxieties of suppliers and managers will be eliminated.

Finally, financial subsidies for shared parking should be supported by the government. As for the results of SEM methodology, the cost risks outweigh the economic benefits in the practice of shared parking. This is also the main reason why shared parking could not be widely applied. Despite this, shared parking could generate more social benefits in theory, such as decreasing the vacant berths and alleviating parking and traffic congestion. The most anxiety-filled questions concerning the suppliers and managers were the cost risks, which decrease the intention to share parking spaces. Therefore, financial subsidies should be supported by the government to alleviate the cost pressures of suppliers and managers. Demonstration projects of shared parking also need government investment.

## 7. Conclusions

The purpose of this paper was to develop and validate the hypothesis that risks, benefits, management pressures, and other latent variables are determinants of shared parking application. Most previous work studies shared parking through modeling the feasibility and parking allocation or evaluating the performance of shared parking. Unlike the existing literature, this study focuses on the reason why shared parking could not be applied widely in China and is intended to establish the relationships between the intentions to share parking spaces and the risks and benefits.

Since the concepts of shared parking is applied as the solution for parking difficulties in many cities in China, unclear risks and benefits are still the biggest cruxes in practice, and now would appear to be a good time to measure suppliers' and managers' assessments of shared parking in order to get an understanding of their acceptance of sharing parking spaces. The current research contributes by identifying key latent variables related to shared parking acceptance and by providing countermeasures aimed at increasing stakeholders' intention to share parking spaces. To summarize the results, six main insights concerning the determinants of shared parking acceptance were found:

- According to the interview, both the suppliers and managers agreed that shared parking provides more available parking resources and alleviates the parking difficulty in residential areas. They could get benefits from the parking charges. But they also worried about the threats of safety and privacy invasion.

- The benefits of shared parking positively and significantly influence the intention of the suppliers and mangers to apply shared parking.
- Conversely, the risks and management pressures negatively significantly affect the sharing willingness of parking suppliers and mangers to apply shared parking.
- Cost risks and security risks are the most important factors for parking suppliers' acceptance intention to share parking and management pressures are the major determinant for the managers' intention to share parking spaces.
- The compensation effect between the risks and benefits of shared parking exists. The greater the benefits of shared parking, the stronger the risk tolerance.
- Social benefits directly determine suppliers' and mangers' acceptance intention of shared parking. Compared to the economic benefits, social benefits are not easy to emerge and are often ignored.

These findings have implications for increasing suppliers' and managers' acceptance intention of shared parking and promoting the applications of shared parking. Some implementation strategies of shared parking were proposed as follows:

- In order to regulate the risks and benefits of shared parking, the new-style cooperative safety supervision mechanism of shared parking should be established by the stakeholders.
- New AIPT should be introduced to reduce the pressures of shared parking management.
- The demonstration project of shared parking should be implemented by the government, supplier, and manager to improve public awareness of social benefits generated by shared parking.
- Shared parking will be considered under the connected and autonomous vehicles environment in the future [33–35].

In order to turn around the situation that the cost risks outweigh the economic benefits in the application of shared parking, financial subsidies should be supported by the government to alleviate the cost pressures of the supplier and manager.

This paper helps the residential parking suppliers and managers to clarify the risks and benefits of participating in shared parking and analyses the impact of risks and benefits on the mode of shared parking from the perspective of parking suppliers and managers. The proposals provide a direction for residential parking suppliers and managers to decide whether to take part in shared parking. If parking suppliers and managers put forward relevant appeals according to the specific situation of residential areas, then a reasonable solution can be obtained according to the proposal in this paper. In order to implement the mode of shared parking, the first step of shared parking is to build an Internet parking service platform such as the shared parking applications "ETCP", "PP parking", "Easy to stop", "Line", "Good parking", and "Meter parking". The parking supplier updates the relevant information of the parking space on the platform in a timely manner and adopts the mode of parking reservation to ensure that the parking suppliers must have a parking space. In addition, shared parking service platforms should be based on LBS (Load Balance System) which can not only recommend and query the surrounding car parks according to the user's geographical location, but also provide the service of parking reservation.

**Author Contributions:** The authors confirm contribution to the paper as follows: study conception and design: J.X., X.Y. (Xiaofei Ye); data collection: X.Y. (Xingchen Yan), Z.Y. (Zhen Yang), T.W.; analysis and interpretation of results: J.X., X.Y. (Xiaofei Ye), Z.Y. (Zhongzhen Yang); draft manuscript preparation: J.X., X.Y. (Xiaofei Ye), L.L. All authors have read and agreed to the published version of the manuscript.

**Funding:** This research was funded by the projects of the National Natural Science Foundation of China Grant number [51408322], Natural Science Foundation of Zhejiang Province, China Grant number [LQ17E080007], Natural Science Foundation of Ningbo, Zhejiang Province, China Grant number [2017A610139], Basic Research Program of Science Grant number [BK20180775] and Technology Commission Foundation of Jiangsu Province, China Grant number [BK20170932].

**Acknowledgments:** The authors would like to thank two anonymous reviewers for constructive comments and suggestions. The authors acknowledge the financial support from the relevant institutions. The authors also thank the respondents for providing data and information that were essential for this work.

**Conflicts of Interest:** The authors declare no conflict of interest.

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
