# Peer review of "Impact of Risk and Benefit on the Suppliers’ and Managers’ Intention of Shared Parking in Residential Areas"

_sustainability, doi:10.3390/su12010268_

Round 1

Reviewer 1 Report

The authors propose a survey analysis for determining user perception in the case of shared parking areas in residential contexts.

First of all, I reckon that the authors should modify the paper title and the abstract which are misleading. Indeed, it seems that they have investigated the effects in terms of road flows, travel demand, parking lot occupancy, parking management revenues, etc., while they have “asked people their opinion”.

Hence, I reckon that authors should:

provide a new and more suitable title provide a new and more suitable abstract where they show the real contents of the paper; provide a considerable extension of the literature in order to improve the placement of their contribution. In particular, I suggest them to investigate (by means of an online search engine such as Google, ResearchGate, etc.) the following topics: the urban parking pricing problem pricing policies in urban contexts user behaviour in the case of different parking schemes effects of cruising for parking on traffic flow provide an analysis of survey techniques (State Preferences vs Revealed Preferences) in order to classify their survey approaches. Moreover, authors should indicate the sampling rate (i.e. the ratio between surveyed people and parking users) in order to clearly reveal the reliability of the obtained results highlight the innovatively of their approaches compares to the existing literature highlight the gap filled by their proposal how their proposal (i.e. the parking share) may be implemented by means of the new technologies provide an overview of cooperative driving, as well as an autonomous vehicle, and analyse effects that these kinds of vehicles may have on the parking sharing.

Reviewer 2 Report

The article refers to a residential area, and the sources cited indicate that the most susceptible to this form of parking use are car parks, e.g. at churches. It's worth commenting.

I had no explanation why Ningbo was used in the article as an example. The authors cite 3 other Chinese agglomerations where such parking is being attempted (Shanghai, Beijing and Guangzhou).

It is worth explaining in the content TOD (Transit Oriented Development).

It should be noted that the research sample (800) was not randomly selected and why it could not be.

The numbering of chapters and subsections should be corrected.

2 x is 4.1.1, and probably it was 4.2.1 and 4.2.2 2 x is chapter 4 (Discussion should be 5, and Conclusions 6, Patents 7)

In references, the authors incorrectly refer to first names instead of the surnames of the authors of individual publications used. This situation occurs in:

(4): Mary is the first name, the surname is Smith, so it should be M.S. Smith (11): Abdul is the first name surname Hamid, so it should be:
A. Hamid (17): Jnga is the first name and surname is Molenda, so it should be I. Molenda (19): Christos is the first and surname of Cassandras, so it should be, C.G. Cassandras

In addition, the authors cite reports to which authors are not assigned (7) (15), citing names not shown anywhere.

The reviewer did not meet the references in the names of the persons who submitted the reports (7). Why in (14)is a disproportionately large font?

References are definitely to be improved, because it is of great importance for the authors of the publications cited!!

Round 2

Reviewer 1 Report

Auhtors have modified the apper according to my observations